# The Role of Endoscopic Ultrasound in Early Chronic Pancreatitis

**DOI:** 10.3390/diagnostics14030298

**Published:** 2024-01-30

**Authors:** Jimil Shah, Abhirup Chatterjee, Truptesh H. Kothari

**Affiliations:** 1Department of Gastroenterology, Postgraduate Institute of Medical Education and Research (PGIMER), Chandigarh 160012, India; shah.jimil@pgimer.edu.in (J.S.); chatterjee.abhirup@pgimer.edu.in (A.C.); 2Division of Gastroenterology and Hepatology, University of Rochester Medical Center, Rochester, NY 14642, USA

**Keywords:** chronic pancreatitis, alcohol, SPINK1, abdominal pain

## Abstract

Chronic pancreatitis (CP) is an irreversible and progressive inflammation of the pancreas that can involve both pancreatic parenchyma and the pancreatic duct. CP results in morphological changes in the gland in the form of fibrosis and calcification along with functional impairment in the form of exocrine and endocrine insufficiency. Studies on the natural history of CP reveal the irreversibility of the condition and the resultant plethora of complications, of which pancreatic adenocarcinoma is the most dreaded one. In Japanese population-based studies by Otsuki and Fuzino et al., CP was clearly shown to reduce lifespan among males and females by 10.5 years and 16 years, respectively. This dismal prognosis is superadded to significant morbidity due to pain and poor quality of life, creating a significant burden on health and health-related infrastructure. These factors have led researchers to conceptualize early CP, which, theoretically, is a reversible stage in the disease spectrum characterised by ongoing pancreatic injury with the presence of clinical symptoms and the absence of classical imaging features of CP. Subsequently, the disease is thought to progress through a compensated stage, a transitional stage, and to culminate in a decompensated stage, with florid evidence of the functional impairment of the gland. In this focused review, we will discuss the definition and concept of early CP, the risk factors and natural history of the development of CP, and the role of various modalities of EUS in the timely diagnosis of early CP.

## 1. Introduction

Chronic pancreatitis (CP) is an irreversible and progressive inflammation of the pancreas that can involve both pancreatic parenchyma and the pancreatic duct. CP results in morphological changes in the gland in the form of fibrosis and calcification along with functional impairment in the form of exocrine and endocrine insufficiency. Studies on the natural history of CP reveal the irreversibility of the condition and the resultant plethora of complications, of which pancreatic adenocarcinoma is the most dreaded one [1,2]. In Japanese population-based studies by Otsuki and Fuzino et al., CP was clearly shown to reduce lifespan among males and females by 10.5 years and 16 years, respectively [3]. Most of these patients had occult pancreatic malignancies which were diagnosed late and were detected to be the commonest cause of mortality among these patients, with a standardized mortality rate of 7.33 [3]. This dismal prognosis is superadded to significant morbidity due to pain and poor quality of life, causing a significant burden on health and health-related infrastructure. These factors have led researchers to conceptualize early CP, which, theoretically, is a reversible stage in the disease spectrum characterised by ongoing pancreatic injury with the presence of clinical symptoms and the absence of classical imaging features of CP. Subsequently, the disease is thought to go through a compensated stage, a transitional stage, and to culminate in a decompensated stage, with florid evidence of the functional impairment of the gland [4]. In this focused review, we will discuss the definition and concept of early CP, the risk factors and natural history of the development of CP, and the role of various modalities of EUS in the timely diagnosis of early CP. 

## 2. The Evolution of the Definition of Early Chronic Pancreatitis (CP)

The term ‘early-stage CP’ was first used by Ammann et al. in 1996 to stratify patients of alcoholic CP before they would develop typical clinical features of CP [5]. The term was introduced as a bridge between alcoholic acute pancreatitis (AP) and CP. American pancreatic Association (APA) and United European Gastroenterology (UEG) evidence-based guidelines on CP were published in 2014 and 2017, respectively. However, none of them could define early CP due to insufficient evidence. Meanwhile, in an international consensus in 2016, Whitcomb et al. proposed a mechanistic definition of CP and adopted the progressive model of CP, characterised by five stages: at risk, acute pancreatitis (AP)-recurrent AP, early CP, established CP, and end-stage CP [6]. Thus, the diagnosis of early CP is conceptualized based on risk factors, inflammatory markers, clinical features, and structural and functional alteration in the gland, and not upon the duration of the disease. 

Initially, attempts to define early CP were based on the morphological changes described as the Rosemont criteria (RC) for CP [7]. Early CP is characterised by features that are described as major B or minor EUS findings for the diagnosis of CP in the Rosemont criteria. This includes parenchymal changes like lobularity, hyperechoic foci in the absence of shadowing, strands, and ductal changes like dilated side branches and a hyperechoic MPD margin [7]. In studies, however, most of the parenchymal changes do not correlate with histological changes, neither in surgically resected specimens nor in EUS-guided fine-needle biopsy samples [8,9]. On the other hand, changes like hyperechoic ductal margin, dilated side branches, and the presence of cysts are shown to correlate histologically. To obviate this contradiction, the Japanese Pancreatic Society (JPS) first proposed a definition of early CP in 2009, which included features of hyperechoic foci without shadowing, lobularity with or without honeycombing, hyperechoic ductal margin, intra-parenchymal cysts, etc. [3]. However, in the international working group and the PancreaFest working group consensus statement, it was recognized that the morphological-based diagnosis of early CP was inaccurate due to low specificity, a lack of histological correlation, an unpredictable clinical course, and overdependence on imaging [6]. This has led to further revision of the JPS 2009 criteria to improve specificity, reduce inter-observer variation, and make it simpler to use. The JPS 2019 definition introduced risk factors as a history of AP, and clinical features in addition to the imaging features. Repeated upper abdominal pain, elevated serum or urinary levels of pancreatic enzymes, impaired pancreatic exocrine function, heavy alcohol intake (>60 g/day of pure ethanol or equivalent), genetic mutations (PRSS1 or SPINK1), and history of AP were included in the clinical diagnostic criteria of early CP. The EUS criteria involved (1) hyperechoic foci (without shadowing) or strands, (2) lobularity, (3) dilated side branches, and (4) a hyperechoic MPD margin. Early CP is defined, according to the JPS 2019 criteria if any three (out of seven) of the clinical criteria and any two (out of four, including one or two) of the imaging criteria are fulfilled. In contrast to the JPS 2009 definition, lobularity with and without honeycombing were merged and cysts as criteria were removed to avoid confusion with the cystic intraductal neoplasm of the pancreas [7]. However, prospective studies evaluating the accuracy of JPS 2019 diagnostic criteria are still lacking. Future studies on this topic should evaluate the validity of the JPS 2019 diagnostic criteria in the diagnosis of early CP, its inter-observer variability, and its impact on the natural history of the development of full-blown CP (Table 1 and Table 2).

## 3. Risk Factors for Early CP

Acute pancreatitis is thought to incite inflammation in the organ, which may eventually resolve, leading to the regeneration of injured cells. However, recurrent attacks of acute pancreatitis (RAP) can predispose one to chronic inflammation and fibrosis. In the consensus by Whitcomb et al., both AP and RAP are considered as major risk factors for CP [6]. In a recent population-based study, three episodes of AP significantly increases risk of CP (0%, 1%, 16%, and 50% probability of CP after first, second, third, and fourth attack of AP, respectively) [11]. In a longitudinal study, cumulative incidence of chronic pancreatitis after an attack of AP was 13% over 10 years and 16% over 20 years. In the same study, incidence of CP was found to hike up sharply to 38% within 2 years after a second attack of AP. The progression of AP to CP was found only among alcoholics, which was independent of the severity of the first attack of AP and the cessation of alcohol intake and smoking [12]. Similar findings were shown in another cohort of 7456 patients with incident AP. In that study, smoking and RAP were significant risk factors for CP development [13]. An increased risk of CP after a third attack of RAP is attributed to the increased prevalence of alcoholism over the biliary etiology of AP, a higher grade of pancreatic injury after a third attack of AP, and increased local and systemic complications after the third attack of AP which might contribute to progression to chronic pancreatic inflammation [12]. Alcohol and smoking are also well-known toxins to the acinar, as well as the ductal cells. Heavy alcohol use can lead to parenchymal changes detectable in EUS, even in the absence of any clinical symptoms [14,15,16,17]. In one study among asymptomatic healthy individuals, 4% had changes in EUS suggestive of CP, leading to further confusion [17]. Though these changes disappear after the stoppage of ongoing alcohol use, it is still not clear whether these changes are indicators of ECP or are non-specific changes. 

Genetic mutation as a risk factor for CP is well established. However, for ECP, the role of specific germline mutations, which are high risk and highly penetrative, becomes relevant, more so when they are present before the clinical manifestation. Multiple molecular pathways are known to interplay in the development of CP, which include (1) the trypsin regulatory pathway, (2) a misfolded protein related to endoplasmic reticulum (ER) stress, and (3) ductal secretion. The importance of the detection of genetic mutation includes determining the etiology of pancreatitis, determining the underlying molecular pathway, identifying pathogenic modifier genes, predicting the course of illness, and developing potential therapeutic targets. However, genetic testing has some limitations, like a lack of counselling protocol for all mutations, the detection of mutations of uncertain significance, the role of testing in asymptomatic individuals, and limited available treatment options even after a diagnosis of the same. At present, genetic testing is not routinely recommended in asymptomatic individuals but the detection of mutations increases the likelihood of a diagnosis of ECP in an appropriate clinical background. 

## 4. Natural History and Progression

There is little available literature on the natural history of early CP, with a significant knowledge gap in the natural history of the progression of CP. In an epidemiological survey among the Japanese population by Masamune et al., authors have shown that only about 5% of patients fulfilling the JPS (2009) criteria for early CP progressed to definite CP after a 2-year follow-up, whereas 62.7% were downgraded. These patients were labelled as ‘possible early CP’, ‘possible CP’, or other diagnoses at the end of two years of follow-ups [10,18]. In a multicentre prospective cohort study among 88 patients diagnosed with ECP according to JPS 2009 criteria, 83 were followed up for 2 years. Among them, all four patients (4.8%) who developed definite CP after 2 years were found to be males and alcoholics with ongoing alcohol intakes. Three of them had previous AP and two of them had an attack of AP during the follow-up. In another prospective study by Ito et al., 113 patients diagnosed with ECP by JPS 2009 criteria were enrolled and 53 were followed up. Nine out of these fifty-two ECP patients (9.6%) were found to progress to established CP at the end of 2 years. On further analysis, all of them were found to be alcoholics and 80% had ongoing alcohol intakes. Interestingly, 61.5% of these patients had to be downgraded to suspicious ECP or had a disappearance of symptoms, and one-third failed to show any change over the follow-up period [6]. In another retrospective study, 118 patients with clinical symptoms of CP, but without definitive imaging or pathological findings of CP, were assessed. Among them, 40 patients showed minimal change features of CP on EUS (labelled the MCEUS group). Over a median follow-up of 30 months, 12 (30%) were found to have progressed to definite CP. Most of these patients (67%) had a history of significant alcohol intake (more than 62 units of alcohol weekly) or smoking (83%), and 58% had past attacks of acute pancreatitis [19]. Thus, the progression of ECP seems to be strongly dependent upon ongoing pancreatic injury, due to risk factors like alcohol, smoking, and repeated attacks of acute pancreatitis (Figure 1). So, the detection of ECP gives a strong pointer towards having a strict abstinence from alcohol and smoking. However, in the absence of disease-modifying agents, whether the early detection of CP can change the natural history of disease and improve the quality of life is yet to be explored in the future prospective studies. 

## 5. The Role of Imaging in ECP: A Diagnostic Conundrum

Traditionally, imaging in CP was dependent upon computed tomography (CT) and magnetic resonance imaging (MRI) with magnetic resonance cholangiopancreatography (MRCP). In a meta-analysis, the detection of CP by CT had a sensitivity and specificity of 75% and 91%, respectively [20]. Both ductal and parenchymal changes in CP can be appreciated in CT with the added advantage of screening for pancreatic cancer, diagnosing collections, excluding alternate diagnosis, and the detection of complications like biliary and vascular complications [21]. Despite this, CT has its limitations in ECP diagnosis due to poor accuracy in diagnosing subtle changes in the pancreas, leading to a high false negative rate [22,23]. A dilated MPD (2–4 mm), pseudocysts, dilated side branches, and mild pancreatomegaly are changes that have been described in studies in ECP [24]. However, the subtle changes that remain undetectable by CT make CT unreliable for ECP diagnosis. 

MRI with MRCP has an advantage over CT for detecting more subtle changes seen in the initial phases of CP (mild changes), and better appreciation of ductal changes like dilatation, strictures, and pathological side branches. In a study by Tirke et al., T1-weighted MRI was found to detect parenchymal changes in CP with a sensitivity and specificity of 77% and 83%, respectively. Newer advances in MR imaging techniques like multiparametric mapping, MR-elastography and diffusion-weighted imaging are shown to improve accuracy and specificity, and are likely to be useful in diagnosing ECP [25]. In a study by Wang et al., multiparametric mapping involving T1, T2, and an apparent diffusion coefficient or ADC has been shown to have better sensitivity and specificity (91% and 85.8%, respectively) [26]. Secretin-enhanced MRI has better accuracy than conventional MRCP for suspected CP patients with negative CT and conventional MR, with the ability to detect ductal compliance, semiquantitative or the quantitative estimation of pancreatic exocrine insufficiency and differentiating CP from small malignancies as the cause of ductal stenosis [21]. Newer sequences like spin labelling and inverse recovery pulse techniques appear promising and can be useful in ECP diagnosis. MRCP criteria for ECP have been described (modified Cambridge criteria) by Schreyer et al., which include MPD dilatation (2–4 mm), pseudocysts ≤1 cm, and irregular MPD with ≥3 pathological side-branches [27]. Moreover, with the widespread availability of MRI, its clinical role soon might be expanded for the diagnosis of ECP. Compared to EUS, MRI is also a non-invasive modality, and has more objective parameters, thus reducing the inter-observer variability. However, studies with head-to-head comparisons of both modalities for the diagnosis of ECP are still lacking. 

Though ERCP has a better ability to delineate subtle ductal changes, it has its drawbacks due to increased complication rate, operator dependence, and chance of falsely detecting ductal changes due to the forceful application of contrast retrogradely across the pancreatic duct. The sole diagnostic role of ERCP in the present era is obsolete, which makes it an invalid option for detecting ECP [24].

## 6. EUS in Early CP

Normal pancreatic parenchyma has a fine reticular pattern on EUS, and MPD appears as a homogenous linear echogenic structure without any prominence or visibility of side branches. In CP, parenchymal fibrosis is represented by an array of findings on EUS (corresponding to certain histopathological findings), including hyperechoic foci (corresponding with focal fibrosis), hyperechoic strands (corresponds with bridging fibrosis), lobularity (interlobular fibrosis), lobular outer margin of the gland (glandular atrophy and fibrosis), and parenchymal calcifications. Ductal changes include dilated MPD, side branch dilatation, intraductal calcification, ductal irregularity (corresponds to focal dilatation or narrowing), and hyperechoic ductal margin (periductal fibrosis) [5,28,29]. Ductal changes are detected early in EUS in the form of mild irregularities in the MPD, the dilatation of the side branches, and the hyperechoic margin of the MPD. Although these findings impart EUS a high sensitivity, there is a high likelihood of false positivity secondary to age-related physiological changes in the pancreatic duct, thereby over-diagnosing ECP. Moreover, being operator-dependent, the findings may also have inter-observer variability, affecting the overall accuracy of EUS in the diagnosis of ECP.

### 6.1. EUS Features of Early CP

JPS (2009) proposed two out of the seven (five parenchymal and two ductal) features to describe imaging features in early CP (Table 3). Subsequently, in the JPS 2019 definition, lobularity with and without honeycombing were merged. Here, we will discuss in detail the various EUS findings described in the definition of ECP and their correlations.
**Lobularity with and without honeycombing:** Lobules are described as well-circumscribed reticulated areas ≥5 mm in size, with a relatively hyperechoic rim compared to the adjacent central area. When these lobules are non-contiguous, the EUS pattern is described as ‘lobularity without honeycombing’. When at least three of such lobules are contiguously located in the body or tail region, the pattern is defined as ‘lobularity with honeycombing’ in EUS [30,31]. (Figure 2) The exact histopathological correlation of lobularity is not precisely known. Studies, however, have demonstrated lobularity to correlate with increased fat and collagen in biopsy specimens, and in a recent study, lobularity was demonstrated to be associated with increased disease severity, a higher level of inflammation, and a trend towards a higher grade of fibrosis and atrophy compared to the absence of lobularity in EUS [32,33];**Hyperechoic foci without shadowing:** Echogenic structures of ≥3 mm in length and width without any posterior acoustic shadowing are defined as ‘hyperechoic foci without shadowing’ in the JPS definition. In standard definition, it is included under ‘hyperechoic foci.’ At least three such foci need to be present to be described as abnormal. The presence of acoustic shadowing signifies calcification. Histologically, they correspond to focal fibrosis (Figure 3);**Stranding:** The presence of hyperechoic lines of ≥3 mm length in a minimum of two directions concerning the plane of imaging is described as ‘stranding’ in the JPS criteria (in standard criteria, it is described as hyperechoic foci with stranding). Abnormal stranding is described when at least three such lines are noted. Stranding corresponds to bridging parenchymal fibrosis in histopathology (Figure 4);**Cysts:** In EUS, they are described as anechoic structures, with/without septations, round or elliptical in shape, measuring ≥2 mm in short axis. Histologically, they correspond to pseudocysts or retention cysts;**Dilatation of the side branches:** It is defined as the presence of ≥3 anechoic, tubular structures communicating with the MPD, each ≥1 mm in width, demonstrable in the body and tail region. Histologically, they correspond to the narrowing of the branch ducts due to micro-fibrosis;**Hyperechoic margin of the MPD:** It is described when the hyperechoic ductal wall over at least 50% of the MPD is demonstrated in the body and the tail of the pancreas. In a linear echoendoscope, MPD assessment on a long axis is difficult. Thus, this finding is often subjective and has low interobserver agreement [34]. Histologically, they correspond to periductal fibrosis. In the study by Sekine et al., the hyperechoic MPD wall was described to correlate with the thinning of the ductal wall on surgical specimens [33] (Figure 2).
diagnostics-14-00298-t003_Table 3Table 3Imaging criteria for ECP and their definitions [3,7,35].CriteriaDefinitionHistopathological AttributesLobularity with honey-combingPresence of ≥3 EUS-defined lobules (reticulated areas surrounded by ≥5 mm rim-like hyperechoic structures) in the body or tail regionInterlobular fibrosisLobularity without honey-combingPresence of non-contiguous lobularityHyperechoic foci without shadowingEchogenic structures ≥3 mm without acoustic shadowFocal fibrosisHyperechoic strandingAt least three hyperechoic lines of ≥3 mm in length in different planes of the imageBridging fibrosisCystsAnechoic structures, with/without septations, round or elliptical in shapePseudocyst or retention cystDilated side branchesPresence of ≥3 tubular anechoic structures arising from the MPD, each ≥1 mm in width, indicative of micro-fibrosisDuctal ectasiaHyperechoic MPD marginEchogenic structure involving at least 50% of the MPDPeriductal fibrosis; thickened ductal wall.


### 6.2. The Correlation between EUS Findings and Histology

B-mode EUS findings have been evaluated in multiple studies to correlate with histological findings. In the study by Varadarajulu et al., certain EUS findings were found to correlate with histology (parenchymal fibrosis, atrophy, and ductal fibrosis) among histologically proven cases of non-calcific chronic pancreatitis. These findings were the presence of foci (*p* < 0.0001), strands (*p* < 0.001), hyperechoic MPD (*p* = 0.03), the irregularity of MPD (*p* < 0.0001), dilated MPD (*p* < 0.0001), and dilated side branches (*p* < 0.001). The study also showed a higher likelihood of CP when diagnosed by EUS based on ductal features compared to parenchymal features [9]. In a more recent study by Sekine et al., an overall good correlation between EUS and histology was found. Two EUS features viz., lobularity and hyperechoic MPD margin correlated well with histopathology. Lobularity correlated significantly with histological atrophy and fibrosis (*p* = 0.0034, and 0.017, respectively), and a hyperechoic ductal margin correlated with the histological thickness of the duct, though this did not reach a statistical level of significance (*p* = 0.06) [33]. Similarly, in a retrospective study, the sensitivity and specificity of EUS for the detection of fibrosis were found to be higher as the disease progressed (84% and 100%, respectively) [36]. Similar to a correlation of EUS findings with histology, the study has also explored the correlation of EUS findings with clinical history and histology. In a retrospective study among 344 patients, individual EUS features and the Rosemont criteria were evaluated with respect to the risk factors for CP including age, smoking, history of AP, and alcohol intake. In a multivariate analysis, lobularity with or without honeycombing was found to strongly correlate with smoking status and previous history of AP. Strong correlations were also found between stranding with alcohol intake and smoking status; hyperechoic foci without shadowing with alcohol and smoking status; dilated side branches with a history of AP; and hyperechoic MPD margin with smoking status. The presence of cysts, however, did not correlate with these risk factors. These three risk factors were also found to positively correlate with Rosemont’s criteria [37].

### 6.3. The Role of EUS Elastography in Early CP

B-mode EUS, although sensitive to CP, is shown to have significant inter-observer variability in studies. In a multicentre study by Koh et al., various EUS features described in CP were evaluated for inter-observer agreement and validation among Asian patients (n = 234). Authors have found the overall kappa score was unsatisfactory (0.54, range 0.14–0.90), and some of the EUS features have a poor agreement, like the hyperechoic duct wall (kappa value 0.14 range 0.01–0.29) [34]. EUS elastography, this plays an important role in the objective assessment of the organ. Elastography is a technique based upon tissue stiffness and deformability, which can be assessed both qualitatively and quantitatively. Qualitative EUS-elastography consists of a color-coded strain distribution map, which is known as an elastogram. In CP, the appearance of the pancreas becomes heterogeneous with green colouration with interspersed blue areas. Semiquantitative elastography is performed using a strain histogram (SH) and strain ratio (SR). Quantitative elastography is shear wave elastography (SWE) [38,39]. Some studies have shown a good correlation between elastographic measurements and stages of CP based on Rosemont’s criteria [40]. Conventional EUS strain elastography is shown to have a sensitivity and specificity of 91% and 91.2%, respectively, for diagnosing CP [39]. However, strain elastography is operator-dependent, and depends on the position and size of the region of interest (ROI). SWE provides a more precise measurement and an absolute value of the stiffness of the pancreatic parenchyma, and has been shown to have excellent sensitivity and specificity (100% and 94%, respectively) for the diagnosis of CP at a cut-off value of 2.19, with an accuracy of 97% in a study by Yamashita et al. SWE measurement was found to have a significant positive correlation with the stage of CP as described in Rosemont’s criteria (r = 0.81), and the median value of SWE for the diagnosis ‘consistent with CP’ and ‘suggestive of CP’ were significantly higher than that of the normal [2.98 (2.29–4.52) and 2.95 (2.19–3.76), respectively, *p*-value < 0.001 for both]. The number of EUS features in the RC was also found to positively correlate with SWE measurement (r = 0.72). (Figure 5). SWE measurements have also been shown to correlate well with exocrine and endocrine insufficiency in CP patients [41]. Despite these promising results, the study failed to show the same performance for indeterminate CP (practically equivalent to ECP), which did not correlate with the RC (median SWE 1.8) and non-CP patients [38]. SWE is also limited due to erroneous measurements among obese patients and due to respiratory movements. Thus, for ECP diagnosis, elastography, though it appears promising, still lacks enough evidence in its favour and needs further studies to explore its role in ECP. One prospective study has also used a combination of B-mode EUS, EUS elastography, the endoscopic pancreatic function test (ePFT), and the secretin-stimulated dynamic assessment of pancreatic duct compliance among patients with suspected ECP based on clinical symptoms and inconclusive EUS findings. About two-thirds (64.1%) of these patients showed positive results for all the four modalities, which makes this a promising approach for detecting ECP on appropriate clinical grounds [42].

## 7. Future Directions in the Diagnosis of ECP

There is still an unmet need in the field of ECP. Considering the complex interplay between genetics, risk factors, varied and nonspecific clinical presentation, the unreliability of cross-sectional imaging for detecting early changes in CP, and the ability of EUS for the same, there seems to be the need for a multi-modality approach for diagnosing ECP with accuracy. Moreover, understanding the natural history of ECP is also of utmost importance for knowing at-risk individuals, where timely intervention can reduce the disease progression. Moreover, studies on therapies to prevent the progression from ECP to calcific or symptomatic pancreatitis are also important.

## Figures and Tables

**Figure 1 diagnostics-14-00298-f001:**
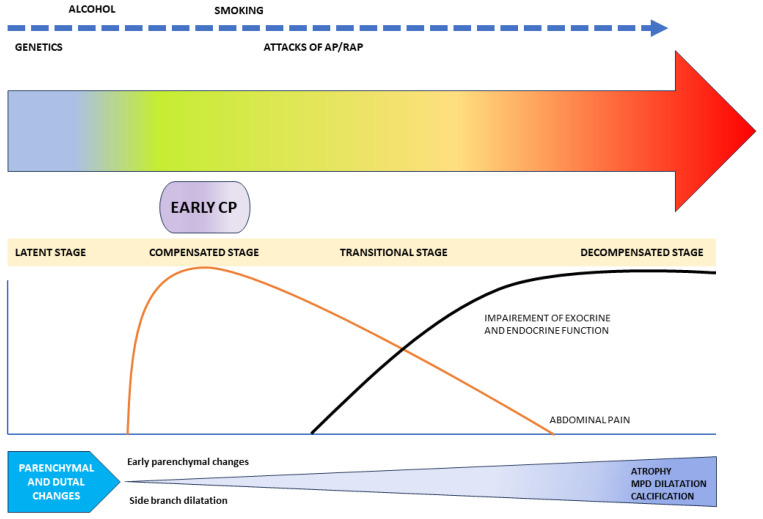
Schematic representation of natural history of chronic pancreatitis.

**Figure 2 diagnostics-14-00298-f002:**
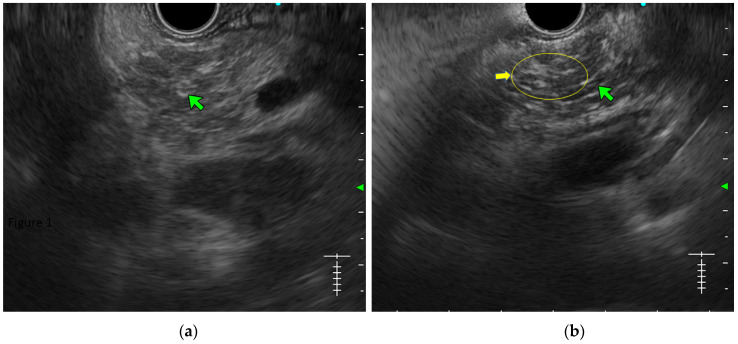
(**a**) Figure showing a lobularity without honeycombing (well-circumscribed reticulated areas, with a relatively hyperechoic rim), (**b**) showing lobularity with honeycombing (three contiguous lobule yellow arrows) along with hyperechoic duct margins (green arrow).

**Figure 3 diagnostics-14-00298-f003:**
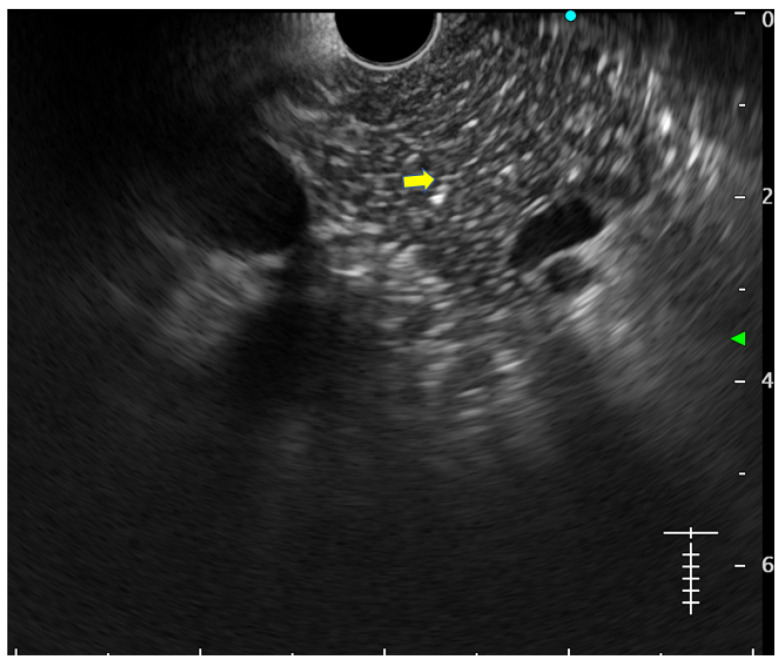
Showing hyperechoic foci without shadowing (yellow arrow).

**Figure 4 diagnostics-14-00298-f004:**
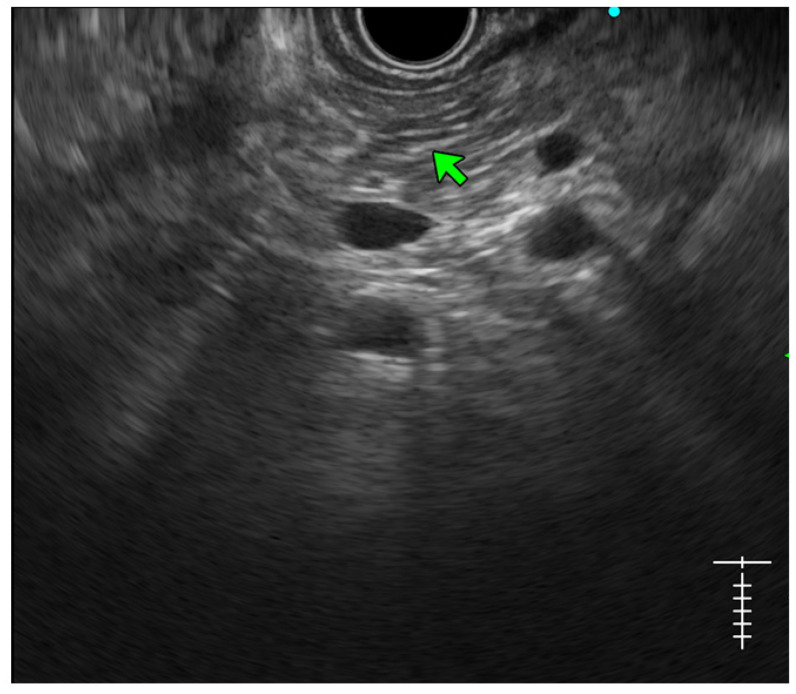
Showing hyperechoic stranding in different planes of images.

**Figure 5 diagnostics-14-00298-f005:**
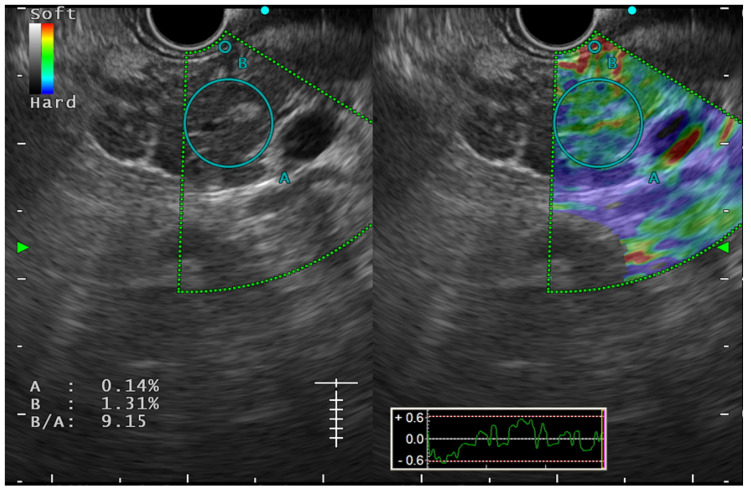
Elastography image showing heterogenous soft to hard pattern on histogram along with strain ratio (B/A) of 9.15, suggestive of chronic pancreatitis.

**Table 1 diagnostics-14-00298-t001:** Japanese Pancreatic Society diagnostic criteria for early chronic pancreatitis (JPS 2009) [3].

**Clinical signs** (1) Repeated upper abdominal pain (2) Abnormal pancreatic enzyme levels in the serum or urine (3) Abnormal pancreatic exocrine function (4) Continuous heavy drinking of alcohol equivalent to or more than 80 g/day of pure ethanol
**Imaging findings (Either a or b)** More than two among the seven features including any of (1)–(4) (1) Lobularity with honeycombing (2) Lobularity without honeycombing (3) Hyperechoic foci without shadowing (4) Stranding (5) Cysts (6) Dilated side branches (7) Hyperechoic main pancreatic duct (MPD) margin Irregular dilatation of more than three duct branches on ERP

**Table 2 diagnostics-14-00298-t002:** Japanese Pancreatic Society diagnostic criteria for early chronic pancreatitis (JPS 2019) [10].

**Clinical features** (1) Repeated epigastric or back pain; (2) Outlier of pancreatic enzyme levels in the serum or urine; (3) Outlier of pancreatic exocrine function; (4) Continuous heavy drinking of alcohol equivalent to or more than 60 g/day of pure ethanol (EtOH 60 g/day), or pancreatitis-related susceptibility genes for the continuous heavy drinking of alcohol; (5) Previous history of acute pancreatitis.
**Imaging findings of early chronic pancreatitis (either a or b)** More than two features among the following four features of EUS findings, including at least one of (1)–(4) (1) Hyperechoic foci; non-shadowing/Stranding; (2) Lobularity [non-honeycombing/honeycombing type]; (3) Hyperechoic main pancreatic duct margin; (4) Dilated side branches; Irregular dilatation of more than three duct branches on ERCP or MRCP findings.

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
