# Peer review of "The Role of Endoscopic Ultrasound in Early Chronic Pancreatitis"

_diagnostics, 2024, doi:10.3390/diagnostics14030298_

Round 1

Reviewer 1 Report

Comments and Suggestions for Authors

This review aims to discuss the use of EUS in the early stages of chronic pancreatitis. The article provides a comprehensive overview of chronic pancreatitis (CP), covering its definition, natural history, risk factors, and the role of endoscopic ultrasound (EUS) in the diagnosis of early CP. The review incorporates relevant scientific studies and population-based research, lending validity to the information presented. This strengthens the article's scientific quality. The article is well-organized with clear section headings, making it easy for readers to navigate through different aspects of CP and early CP diagnosis. Figures and tables enhance the topic. The article also discusses the evolution of the definition of early CP, providing a historical context that helps readers understand diagnostic criteria progress.

I found some minor critical points. By addressing the following 5 points, the review can be improved in terms of educational value and scientific quality.

1. Prospective studies: The article mentions that prospective studies evaluating the accuracy of the JPS 2019 diagnostic criteria and their impact on natural history are lacking. Suggesting with more emphasis the need for and encouraging future prospective studies in this area would strengthen the critical evaluation.

2. Elaboration on limitations: while the article touches on the limitations of EUS and imaging modalities, a more in-depth discussion on the limitations and challenges faced in diagnosing early CP, such as false positives and operator dependence, would enhance the critical analysis.

3. Incorporation of opposing views if possible and if present in literature : If there are differences of opinions or conflicting evidence in the field of early CP diagnosis, it would be beneficial and would help readers understand the nuances and ongoing debates.

4. Patient outcomes : The article briefly mentions the natural history of early CP but could benefit from a more extensive discussion on patient outcomes, prognosis, and the impact of early diagnosis on treatment and quality of life as well as the risk of cancers during patient follow up.

5. Clarity in language: Some sentences are complex: simplify them for a broader audience; it will enhance the educational value of the article.

Comments on the Quality of English Language

Clarity in language: Some sentences are complex: simplify them for a broader audience; it will enhance the educational value of the article.

Author Response

Added 

Reviewer 2 Report

Comments and Suggestions for Authors

l   The article underscores the evolving nature of the definition of ECP and the challenges in its diagnosis. It emphasizes the potential of EUS, particularly elastography, in enhancing diagnostic accuracy.

l   However, since MR imaging is non-invasive and newer advances of this technique showed promising results, the aspect of comparison between EUS and MRI for diagnosis of ECP should be addressed more in detail.

Author Response

Added 

Round 2

Reviewer 2 Report

Comments and Suggestions for Authors

Since the revised version of this manuscript has made significant modification, I am fine to recommend the acceptance for publication.